

# Effects of elevated $CO_2$ on predator avoidance behaviour by reef fishes is not altered by experimental test water

Philip L. Munday[1], Megan J. Welch[1,2], Bridie J.M. Allan[1,2], Sue-Ann Watson[1], Shannon J. McMahon[1,2] and Mark I. McCormick[1,2]

[1] ARC Centre of Excellence for Coral Reef Studies, James Cook University, Townsville, Queensland, Australia
[2] College of Marine and Environmental Science, James Cook University, Townsville, Queensland, Australia

## ABSTRACT

Pioneering studies into the effects of elevated $CO_2$ on the behaviour of reef fishes often tested high-$CO_2$ reared fish using control water in the test arena. While subsequent studies using rearing treatment water (control or high $CO_2$) in the test arena have confirmed the effects of high $CO_2$ on a range of reef fish behaviours, a further investigation into the use of different test water in the experimental arena is warranted. Here, we used a fully factorial design to test the effect of rearing treatment water (control or high $CO_2$) and experimental test water (control or high $CO_2$) on antipredator responses of larval reef fishes. We tested antipredator behaviour in larval clownfish *Amphiprion percula* and ambon damselfish *Pomacentrus amboinensis*, two species that have been used in previous high CO2 experiments. Specifically, we tested if: (1) using control or high $CO_2$ water in a two channel flume influenced the response of larval clownfish to predator odour; and (2) using control or high $CO_2$ water in the test arena influenced the escape response of larval damselfish to a startle stimulus. Finally, (3) because the effects of high $CO_2$ on fish behaviour appear to be caused by altered function of the GABA-A neurotransmitter we tested if antipredator behaviours were restored in clownfish treated with a GABA antagonist (gabazine) in high $CO_2$ water. Larval clownfish reared from hatching in control water (496 μatm) strongly avoided predator cue whereas larval clownfish reared from hatching in high $CO_2$ (1,022 μatm) were attracted to the predator cue, as has been reported in previous studies. There was no effect on fish responses of using either control or high $CO_2$ water in the flume. Larval damselfish reared for four days in high $CO_2$ (1,051 μatm) exhibited a slower response to a startle stimulus and slower escape speed compared with fish reared in control conditions (464 μatm). There was no effect of test water on escape responses. Treatment of high-$CO_2$ reared clownfish with 4 mg l$^{-1}$ gabazine in high $CO_2$ seawater restored the normal response to predator odour, as has been previously reported with fish tested in control water. Our results show that using control water in the experimental trials did not influence the results of previous studies on antipredator behaviour of reef fishes and also supports the results of novel experiments conducted in natural reef habitat at ambient $CO_2$ levels.

Corresponding author
Philip L. Munday,
philip.munday@jcu.edu.au

## INTRODUCTION

Rising concentrations of atmospheric carbon dioxide ($CO_2$) have caused an increased uptake of $CO_2$ by the ocean, leading to a decline in seawater pH and changes in the relative concentration of carbonate and bicarbonate ions, a process called ocean acidification (*Sabine et al., 2004*; *Doney, 2010*). The partial pressure of carbon dioxide ($pCO_2$) at the ocean surface is increasing at the same rate at the atmosphere (*Doney, 2010*) and thus marine species will need to deal with rising $CO_2$ levels in addition to declining pH and other changes in ocean chemistry (*Portner, Langenbuch & Reipschlager, 2004*). Recent models suggest an amplification of seasonal cycles in ocean $pCO_2$ as atmospheric $CO_2$ continues to rise, such that surface ocean $pCO_2$ will reach 1,000 µatm in summer for atmospheric $CO_2$ concentrations that exceed 650 parts per million. Hypercapnic conditions (>1,000 µatm $CO_2$) are projected to occur in up to half the surface ocean by 2100 on the current $CO_2$ emissions trajectory (*McNeil & Sasse, 2016*).

Elevated $CO_2$ levels can have a variety of effects on the physiology, life history and behaviour of marine organisms (*Portner, Langenbuch & Reipschlager, 2004*). Recent studies have demonstrated that high $CO_2$ levels can fundamentally alter the behaviour of marine fishes and some invertebrates (reviewed by *Briffa et al., 2012*; *Clements & Hunt, 2015*; *Heuer & Grosell, 2014*; *Nagelkerken & Munday, 2016*). Of particular concern is an impaired response to the threat of predation, because it may increase mortality rates in natural populations (*Munday et al., 2010*; *Ferrari et al., 2011a*; *Chivers et al., 2014*). Larval and juvenile reef fish can exhibit an innate aversion to chemical cues released by predators and chemical alarm cues from injured conspecifics, because they indicate a heightened risk of predation (*Holmes & McCormick, 2010*; *Dixson, Pratchett & Munday, 2012*). Individuals typically respond to these chemical cues by reducing activity and seeking shelter (*Ferrari, Wisenden & Chivers, 2010*). However, larval reef fish that have been reared at near-future $CO_2$ levels do not reduce activity or stop feeding in the presence of chemical alarm cues (*Ferrari et al., 2011a*; *Lönnstedt et al., 2013*) and even become attracted to chemical cues from a predator (called predator odour from herein) and chemical alarm cues when presented in a two channel flume (*Dixson, Munday & Jones, 2010*; *Welch et al., 2014*). Furthermore, juvenile fish exposed to high $CO_2$ lose their ability for associative learning (*Ferrari et al., 2012*), a process that enables fine-tuning of risk assessment by the association of chemical alarms cues with the identity of specific predators. Finally, high $CO_2$ affects the kinematics of predator–prey interactions, with juvenile prey exposed to elevated $CO_2$ allowing predators to get closer before responding (*Allan et al., 2013*) and exhibiting reduced escape speeds and distances compared with fish reared at current-day $CO_2$ levels (*Allan et al., 2013*; *Allan et al., 2014*). These changes in behaviour alter the outcome of predator–prey interactions, leading to significantly increased rates of mortality of small juveniles in mesocosm experiments (*Ferrari et al., 2011b*) and in fish transplanted to natural coral reef habitat (*Munday et al., 2010*; *Munday et al., 2012*; *Chivers et al., 2014*).

Many of the pioneering studies on the effects of elevated $CO_2$ on reef fish behaviour (e.g., *Munday et al., 2009*; *Dixson, Munday & Jones, 2010*; *Ferrari et al., 2011a*; *Ferrari et al.,2011b*; *Cripps, Munday & McCormick, 2011*) involved rearing fishes in control and

elevated $CO_2$ conditions and then testing their responses to chemical cues in control water only. This method was chosen in these early experiments owing to logistical constraints and because pilot experiments showed that the response of high-$CO_2$ exposed fish to predator cue did not differ if the cue was presented in either control or high $CO_2$ water (*Dixson, Munday & Jones, 2010*; *Munday et al., 2010*). Furthermore, studies with freshwater fishes had shown that a pH reduction of 0.5 units in freshwater (<pH 6.5) irreversibly alters the structure of chemical alarm cues and can dramatically affect prey antipredator responses (*Brown et al., 2002*; *Leduc et al., 2004*; *Leduc, Kelly & Brown, 2004*; reviewed in *Leduc et al., 2013*). Consequently, there was concern that testing marine fish in $CO_2$-acidified water (albeit still above pH 7.5) could confound the interpretation of any effects of high $CO_2$ on the behavioural response of the fish with a diminished efficacy of the chemical cue itself. Testing in control water prevented this potential problem. Finally, small-scale laboratory experiments often have limited ecological relevance and it is challenging to extrapolate their findings to natural communities and ecologically relevant spatial scales (*Nagelkerken & Munday, 2016*). Consequently, it was critical to test the effects of exposure to high $CO_2$ in natural habitats in the field. This involved transplanting larval fish that had been reared in either control or high $CO_2$ seawater back into natural coral reef habitat and monitoring their behaviour (*Munday et al., 2010*; *Munday et al., 2012*; *McCormick, Watson & Munday, 2013*).

Since these initial studies, a number of other experiments have been conducted in which the respective treatment seawater in which the fish were reared (control or high $CO_2$) has been used in the experimental trials, confirming the results of earlier studies (e.g., *Allan et al., 2014*; *Dixson et al., 2015*; *Welch et al., 2014*; *Nagelkerken et al., 2016*). Nevertheless, a more thorough validation study of the test water used in previous antipredator experiments is warranted. Here, we compared the use of control versus high $CO_2$ seawater (called test water from hereafter) in experimental trials designed to test the antipredator responses of larval reef fishes reared in high $CO_2$. We conducted antipredator experiments for two species that have been widely used in previous high $CO_2$ experiments, the clownfish *Amphiprion percula* and the ambon damselfish *Pomacentrus amboinensis*. Specifically, we tested if: (1) using control or high $CO_2$ water in a two channel flume influenced the response of larval clownfish to predator odour in clownfish reared in either control or high $CO_2$; and (2) using control or high $CO_2$ water in the test arena influenced the kinematic response of larval damselfish to a startle stimulus in fish reared in either control or high $CO_2$. Finally, (3) because the effects of high $CO_2$ on fish behaviour appear to be due to altered function of GABA-A neurotransmitter receptors (*Nilsson et al., 2012*; *Hamilton, Holcombe & Tresguerres, 2014*; *Heuer & Grosell, 2014*) we tested if antipredator behaviours were restored in clownfish treated with a GABA antagonist (gabazine) in high $CO_2$ water. Previous studies have demonstrated that abnormal antipredator behaviour of clownfish and damselfish reared in high $CO_2$ conditions is reversed following treatment with gabazine (*Nilsson et al., 2012*; *Chivers et al., 2014*). However, these previous studies involved high $CO_2$ fish that were treated with gabazine in control water and then behaviourally tested in control water. In this experiment we therefore conducted the same gabazine treatments as

previously used, except the drug was administered and behaviour was tested in high $CO_2$ water.

## MATERIALS AND METHODS

The experiment with larval clownfish was conducted at James Cook University's (JCU) experimental aquarium facility in Townsville, Australia, where previous high $CO_2$ experiments with this species have been conducted. The experiment with larval ambon damselfish was conducted at Lizard Island Research Station (LIRS) on the northern Great Barrier Reef, where previous high $CO_2$ experiments with this species have been conducted. Both experiments were conducted during February 2016. Because our goal was to assess the use of control seawater in experimental trials of previous experiments (e.g., *Dixson, Munday & Jones, 2010*; *Munday et al., 2010*; *Munday et al., 2012*) we used the same methods as previous experiments, except that we applied a fully factorial design where fish reared in control and high $CO_2$ conditions were behaviourally tested in both control and high $CO_2$ water. Control water was the ambient in the experimental facility at JCU (496 ± 55 S.D. µatm $CO_2$) and at LIRS where water is pumped directly from the Lizard Island lagoon (464 ± 32 µatm $CO_2$). The high $CO_2$ conditions (1,022 ± 37 and 1,051 ± 18 µatm $CO_2$, respectively) matched projections for $CO_2$ levels in open ocean by the end of this century (*McNeil & Sasse, 2016*).

### Response to predator cue

Two clutches of larval clownfish from different parental pairs maintained in ambient seawater conditions at JCU were reared in control and high $CO_2$ conditions from hatching until settlement (12 days post hatching) using standard practices (*Wittenrich, 2007*). Briefly, larvae were reared in 100 l incubator tanks supplied with a continuous flow of treatment seawater. Water temperature was maintained at 29 °C. Photoperiod was 12 hours light and 12 hours dark. A non-viable microalgae blend (Nano 3600–Reed Mariculture[TM]) was drip-fed into the incubator tanks during daylight hours to reduce light levels and enhance larval feeding during the first four days. Larvae were fed live rotifers (15 per ml) for the first four days and then transitioned onto *Artemia* (5 per ml). Fish were not fed on the morning of testing. Each clutch was divided at hatching so that half the clutch was reared in control seawater (496 ± 55 S.D. µatm $CO_2$) and half reared in the high $CO_2$ treatment (1,022 ± 37 S.D. µatm $CO_2$) for 12–15 days. Two 10,000 L recirculating aquarium systems supplied seawater to the incubator tanks, one supplied control seawater and the other high $CO_2$ seawater. To achieve the desired $CO_2$ level in the high $CO_2$ treatment, seawater was dosed with $CO_2$ to a set pH in a 3,000 L sump using an Aqua Medic AT Control System (Aqua Medic, Bissendorf, Germany). The $pCO_2$ in rearing tanks was measured by nondispersive infrared (NDIR) following the method described by *Hari et al. (2008)*. Air in a closed loop was circulated by a small pump (1 l min$^{-1}$ flow rate) through a coil of thin-walled (membrane thickness 0.4 mm, outer ø 3.8 mm) medical grade silicone tubing placed in the tank. $CO_2$ in the closed loop was then measured at 1 min intervals with a Vaisala GMP343 infrared $CO_2$ probe (accuracy ± 5 ppm $CO_2$ + 2% of reading over the range of experimental manipulations). Each reading ($N = 9$ in control and $N = 9$ in

high $CO_2$) lasted 1 hour to ensure complete equilibration of $CO_2$ between tank seawater and the closed loop of air. The average $CO_2$ for the final ten minutes was calculated for each reading. Due to the short duration of the experiment, and because we were primarily concerned about quantifying $pCO_2$, we did not perform alkalinity titrations to paramterize other components of the carbonate system.

The response of settlement stage clownfish (12–15 days old) to predator cue was tested in a two-channel flume (13 cm × 4 cm internal dimensions to the taper) specifically designed for testing the preferences of larval fish to different chemical cues (*Gerlach et al., 2007*). Larvae were given the choice between a stream of seawater containing predator cue versus a stream of seawater without the predator cue. Both streams of water were either control or high $CO_2$ test water for a given trial. Water from the respective paired sources was gravity fed into the flume at 100 ml $min^{-1}$, maintained by a flow meter. Water was exchanged after every fish, with random alternation between the use of control and high $CO_2$ test water in the flume. A dye test was conducted with every water exchange to ensure laminar flow and no mixing of the water streams. For each trial, a single fish was placed in the centre of the downstream end of the flume and allotted a two-minute acclimation period. The position of the fish at 5-second intervals was recorded for the next two minutes. A one-minute rest period followed, during which the water sources were switched to eliminate the effect of any potential side preference by the fish. After the water switch, the fish was re-centred in the downstream end of the flume and the acclimation (2 min) and test period (2 min) were repeated with the predator water on the opposite side of the flume. The time spent in the water stream containing the predator cue was summed for the two 2-minute test periods for each fish. The temperature of the test water was kept within 1 °C of the water temperature in the rearing tanks. Three trials where the difference in water temperature between the flume and the rearing tanks exceeded 1 °C were discarded to avoid any effect of temperature shock on behaviour. The experimenter was blinded to the rearing treatment of the fish, but it was not logistically possible for the experimenter to also be blinded to the water treatments in the flume.

Predator cues were obtained from the common coral-cod, *Cephalopholis miniatus*. Five cod collected from the Great Barrier Reef (supplied by Cairns Marine, Cairns QLD) were maintained in 60 L aquaria supplied with a continuous flow of seawater from a different seawater system to that used to rear the clownfish. Each cod was fed one cube of commercial fish food (Marine Food, Fish Fuel Co$^{TM}$) each evening. Water temperature was maintained at 29 °C. To obtain predator cues for the experiment, the water was drained from two predator tanks and refilled with water from the same system used to rear the clownfish. To match the methods used in previous experiments, water to the predator tank was turned off and predator cue water was collected after a 2 hour soak time. Predator tanks were aerated to maintain oxygen content and water-bathed to maintain a stable temperature. One of the two predator tanks was dosed with $CO_2$ at the end of the 2 hour soak period, immediately before cue water was extracted. $CO_2$ was dosed at low pressure by hand, with gentle stirring, until the pH matched that of the high $CO_2$ rearing treatment. The other predator tank was also gently stirred but was not dosed with $CO_2$. The individual cod used for predator cues was alternated within and between days. An identical procedure was

followed with two tanks that did not house a predator to obtain seawater without predator cue to use in the flume.

### Gabazine treatment

The gabazine treatment followed the flume methodology described above, except that fish were treated with 4 mg l$^{-1}$ gabazine for 30 min immediately before testing in the flume. Eight larval clownfish were randomly selected from the high $CO_2$ treatments and placed in 40 ml of high $CO_2$ water containing gabazine. The fish were gently removed from the gabazine after 30 min and transferred to the flume using a small net. Their response to predator odour was tested in high $CO_2$ water only. Previous experiments demonstrating that gabazine restores antipredator responses in larval fishes reared in high $CO_2$ water have been conducted in control water (*Nilsson et al., 2012*; *Chivers et al., 2014*). This experiment only tested if a similar effect was observed when gabazine was administered and fish tested in high $CO_2$ water.

## Kinematics of escape response

Settlement stage larval *Pomacentrus amboinensis* were collected using light traps in the Lizard Island lagoon and transferred to an environmentally-controlled aquarium facility at LIRS. Settlement stage *P. amboinensis* at Lizard Island have an average age of 15–23 days old (*Kerrigan, 1996*). Larval fish were assigned randomly to four replicate control (464 ± 32 S.D. μatm $CO_2$) or four replicate high-$CO_2$ (1,051 ± 18 S.D. μatm $CO_2$) aquaria. Twenty five *P. amboinensis* were housed in each 32 l (38L × 28W × 30H cm) aquarium. Water temperature was maintained at approximately 29.5 °C. Fish were kept for 4 days in treatment conditions and fed *Artemia* sp. twice daily *ad libitum*. Fish were not fed on the morning of testing. Each aquarium was supplied with control or elevated-$CO_2$ seawater at 750 ml min$^{-1}$. Elevated-$CO_2$ seawater was achieved by dosing with $CO_2$ to a set pH, following standard techniques (*Gattuso et al., 2010*). Seawater was pumped from the ocean into 32 l header tanks where it was diffused with ambient air (control) or 100% $CO_2$ to achieve the desired pH (elevated-$CO_2$ treatment). A pH-controller (Aqua Medic, Germany) attached to the $CO_2$ treatment header tank maintained pH at the desired level. Duplicate control and high $CO_2$ systems supplied seawater to two replicate aquaria in each system. The p$CO_2$ in rearing tanks was measured by NDIR as described above ($N = 9$ in control and $N = 5$ in high $CO_2$).

After rearing for four days in control or high $CO_2$ conditions, the escape response of the damselfish was tested in both control and high $CO_2$ water (i.e., a fully crossed design). Individual fish were placed into the testing arena (Fig. S1), which consisted of a transparent circular acrylic arena (ø 200 mm) within a larger plastic tank (585 × 420 × 330 mm; 60 L) filled to a depth of 100 mm with either control or high $CO_2$ water. A shallow water depth was used to limit vertical displacement of the fish during escape response trials. Water temperature in the experimental arena was 29 °C–30 °C and the water treatment used was alternated after every 2nd trial. The arena was illuminated by LED strip lighting (750 lumens) placed above the water surface on the outside of the tank. Five minutes after being released into the testing arena, an escape response was elicited by the release of a

tapered metal weight from above the water surface. This was accomplished by turning off an electromagnet to which the metal weight was attached. A length of fishing line was attached to the weight so that it stopped when the tapered tip of the weight only just touched the surface of the water after release from the magnet. In order to provide a sudden stimulation and allow quantification of escape latency, the stimulus was released through a white PVC tube (ø 40 mm, length 550 mm) suspended above the experimental arena, with the bottom edge of the pipe at a distance of 10 mm above the water level. The tube hides the weight from view so there is no visual stimulation before the mechanical stimulation occurs. Fish were startled when they moved to the middle portion of the tank. Escape responses were recorded at 480 frames per second (Casio EX-ZR1000) as a silhouette from below obtained through pointing the camera at a mirror angled at 45°. To minimise visual disturbances, black sheeting surrounded the front of the mirror so that any movement within the room was undetected by the fish. Treatment water in the testing arena was changed every 2nd trial to minimise $CO_2$ loss in the high $CO_2$ treatment water. The rate of $CO_2$ loss from the testing arena was found to be negligible within this time frame. The escape response trials were conducted over two consecutive days. The experimenter was blinded to the rearing treatment of the fish when conducting the trials.

From the videos, we quantified latency to initiate an escape, escape distance, escape speed and maximum speed. A 1 cm line was drawn in the centre of the inner arena to enable distance calibration for video analysis. The moment the stimulus weight hit the water was benchmarked by the first detectable water disturbance in the video. The distance of each fish from the point the stimulus first hit the water was also quantified from the video. All videos were analysed with the observer blinded to the treatments.

### Kinematic variables

Kinematic variables associated with the fast-start response were analysed using the image-analysis software Image-J, with a manual tracking plug-in. The centre of mass (CoM) of each fish was tracked for the duration of the response. The following kinematic variables were measured:

(1) Latency to respond (s) was measured as the time interval between the stimulus onset and the first detectable movement leading to the escape of the animal.

(2) Escape distance (m) is a measure of the total distance covered by the fish during the first two flips of the tail (the first two axial bends, i.e., stages 1 and 2 defined based on *Domenici & Blake (1997)*, which is the period considered crucial for avoiding ambush predator attacks (*Webb, 1976*).

(3) Escape speed (m s$^{-1}$) was measured as the distance covered within a fixed time (first 24 ms after initial response) which corresponds to the average duration of the first two tail flips (the first two axial bends, i.e., stages 1 and 2 based on (*Domenici & Blake, 1997*). This period is considered crucial for avoiding predator ambush attacks (*Webb, 1976*).

(4) Maximum speed (m s$^{-1}$) was measured as the maximum speed achieved at any time during stage 1 and stage 2.

## DATA ANALYSIS

Two-way ANOVA was used to compare percent time that larval clownfish spent in predator cue in the flume. Percentage data was logit-transformed for analysis (*Warton & Hui, 2011*). Treatment water (control or high $CO_2$) and test water (control or high $CO_2$) were the fixed factors. ANOVA was conducted with Statistica (version 13). For the gabazine experiment, the percentage time that larval clownfish treated with gabazine spent in predator cue in the flume was compared with control fish using a *t*-test on logit-transformed data. Linear mixed effects models (LME) were used to compare escape responses of larval damselfish. Latency, escape distance, escape speed and maximum speed were each tested separately. Treatment water (control or high $CO_2$), test water (control or high $CO_2$) and day (1 or 2) were fixed factors in the model. The distance of each fish from the point the stimulus hit the water was also included as a fixed effect in the model to account for any variation associated with differences in the location of fish in the chamber when the stimulus occurred. Tank was included in each model as a random effect to account for the subsampling of fish from replicate tanks within each $CO_2$ treatment. Best-fit model selection based on AIC values was automated using 'MuMIn' (*Bartoň, 2015*) within the 'nlme' package in the R software package (*Pinheiro et al., 2013*). The normality, linearity and homoscedasticity of residuals of the models were verified by visual inspection of residual-fit plots. Latency data was log10 transformed before analysis to improve the distribution of the data and a heterogeneous variance structure was used in the model for latency (but not other variables). R code is available in Supplementary Information 1.

This research was conducted in accordance with James Cook University animal ethics guidelines and permits A1961, A1973 and A2080. Fish were collected under permit G12/35117.1 from the Great Barrier Reef Marine Park Authority.

## RESULTS

There was a highly significant effect of treatment water on the percentage time that larval clownfish spent in the water stream containing predator odour ($F_{1,57} = 125.91, p < 0.001$), but no effect of test water ($F_{1,57} = 0.20, p = 0.66$) and no interaction ($F_{1,57} = 0.06, p = 0.81$). Laval clownfish reared in control water strongly avoided predator odour, spending <9% of the time in that water stream (Fig. 1). In contrast, larval clownfish reared in high $CO_2$ water were strongly attracted to predator odour, spending nearly 80% of the time in that water stream (Fig. 1).

As with the response to predator odour, treatment water had a significant effect on escape responses of larval damselfish, and there was no effect of test water. Distance to stimulus did not affect any of the response variables and was not retained in any of the final models. Latency to respond significantly increased in fish reared in elevated $CO_2$ ($F_{1,6} = 7.76, p = 0.032$), with fish displaying latencies that were 30% longer compared with fish reared in control water (Fig.2A). There was no effect of test water or day of testing on latency to respond and these factors were not retained in the final model. Escape distance following stimulation was not significantly different for fish reared in high $CO_2$ compared with control water ($F_{1,5} = 6.15, p = 0.056$) although there was a clear tendency for a shorter

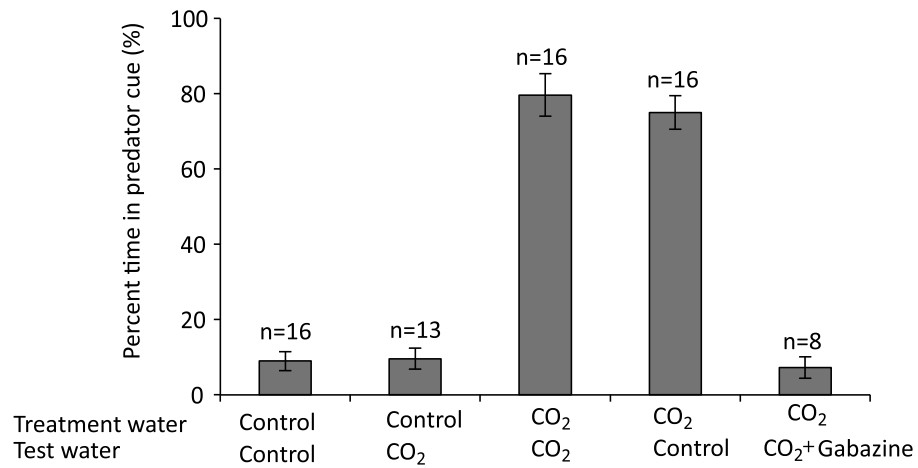

**Figure 1** The effect of rearing treatment water (control or high $CO_2$) and test water used in a two-channel flume (control or high $CO_2$) on the olfactory response of larval *A. percula* to predator cue. One stream of water in the flume contained seawater with predator cue and the other stream of water had untreated seawater. Shown is the percent time (mean $\pm$ s.e.) that fish from each experimental combination spent in the water stream containing predator cue. The final group shows fish that were treated with 4 mg $l^{-1}$ gabazine in high $CO_2$ seawater for 30 minutes before testing in the flume. Sample size shown above the bars.

escape distance in high $CO_2$ fish (Fig. 2B). Day was retained in the final model, but it was not statistically significant ($F_{1,5} = 5.15, p = 0.07$). There was no effect of test water on escape distance and it was not retained in the final model. Escape speed and maximum speed responded similarly with both kinematic variables declining significantly in fish reared in high $CO_2$ compared to fish reared in control water (Fig. 2C; $F_{1,5} = 6.81, p = 0.048$ and Fig. 2D; $F_{1,6} = 9.016, p = 0.0239$). Day was retained in the final model for escape speed, but it was not statistically significant ($F_{1,5} = 3.64, p = 0.115$). Day was not retained in the final model for maximum speed. Again, there was no effect of test water for either escape speed of maximum speed and it was not retained in the final model.

Gabazine reversed the effects of rearing in high $CO_2$ (Fig. 1). Fish reared in high $CO_2$ and then treated with gabazine exhibited a strong avoidance of predator cue, with the response almost identical to fish reared and tested in control water ($t_{22} = 0.04, p = 0.96$).

## DISCUSSION

Previous studies have demonstrated that $CO_2$ levels projected to occur in the surface ocean by the end of this century (700–1,000 µatm) impair predator avoidance behaviours in larval reef fishes (reviewed in *Nagelkerken & Munday, 2016*). However, some of these previous studies have tested the behaviour of high-$CO_2$ reared fish using control water in the experimental trials, which could potentially influence the results if there is an immediate physiological response associated with the transfer of fish from high $CO_2$ rearing water to control test water. Here we show that the results of these earlier studies are robust and that antipredator behaviour is not affected by using control water in the experimental trials. Larval fish reared in high $CO_2$ exhibited the same responses to predator odour in a two channel flume and the same kinematic responses to a startle stimulus regardless of

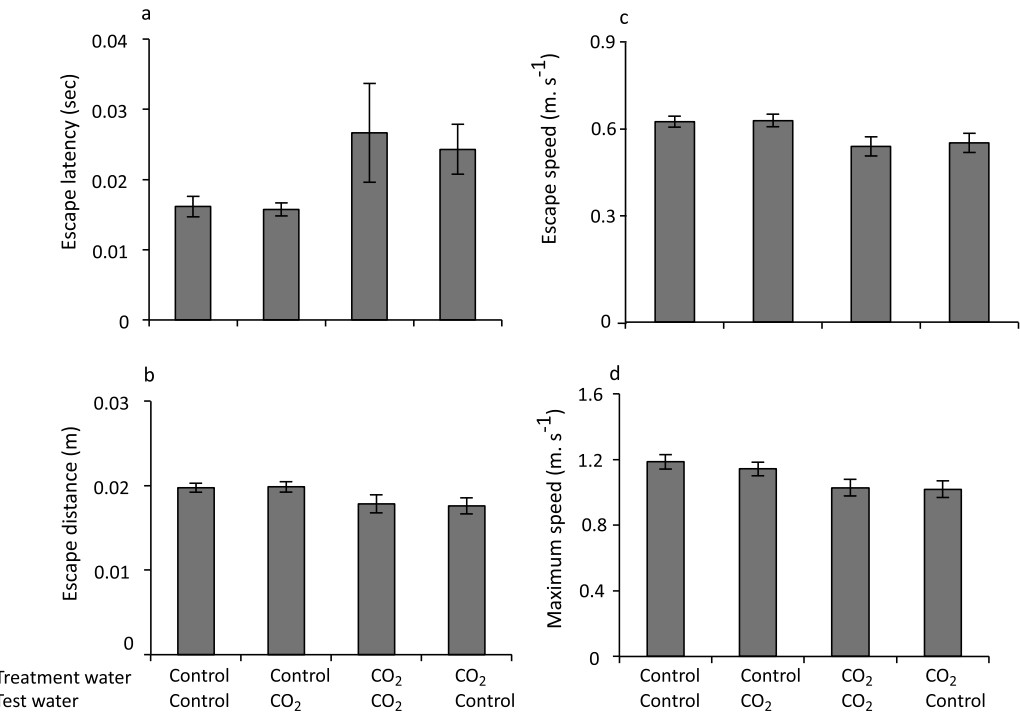

**Figure 2** **The effect of rearing treatment (control or high $CO_2$) and test water used in the experimental trial (control or high $CO_2$) on the escape performance of larval *P. amboinensis*.** Variables displayed are: (A) latency, (B) distance, (C) mean speed, and (D) maximum speed. Errors are standard errors. $N = 26, 28, 26, 28$ for control - control, control - high $CO_2$, high $CO_2$ - high $CO_2$ and high $CO_2$ - control, respectively.

whether high $CO_2$ or control water was used in the experimental trial. Similarly, there was no difference in the behavioural response of fish reared in control seawater when tested in either high $CO_2$ or control water. These results demonstrate that the choice of test water has not influenced the results of previous laboratory studies into the effects of high $CO_2$ on antipredator behaviour.

Our results are consistent with preliminary observations carried out in previous studies showing that the test water (control or high $CO_2$) used in a two-channel flume did not alter the behavioural response of larval fishes to predator odour (*Dixson, Munday & Jones, 2010*; *Munday et al., 2010*). Furthermore, our study extends these findings to the kinematics of predator avoidance, by showing that the type of water used in the test arena does not affect vital components of the escape response, including latency to respond, escape speed and maximum speed. Together these results show that antipredator behaviours of larval fishes that have been reared in high $CO_2$ are not compromised by a return to control seawater for experimental testing. These findings are also relevant to field-based studies that have tested the effects of high $CO_2$ on reef fishes in their natural habitat (e.g., *Munday et al., 2010*; *Munday et al., 2012*; *Devine, Munday & Jones, 2012*; *Devine & Munday, 2013*; *Ferrari et al., 2011a*; *McCormick, Watson & Munday, 2013*). Such studies are critical for predicting the impacts of high $CO_2$ on reef fish populations; however, they involve rearing larval fish for 4–5 days in high $CO_2$ conditions before transferring them to ambient $CO_2$ conditions
in the field. These studies have demonstrated mortality rates from predation are higher in juvenile fish that have been exposed to high $CO_2$ compared with fish exposed to ambient control conditions (*Munday et al., 2010*; *Munday et al., 2012*; *Ferrari et al., 2011a*). Our results indicate that such studies are unlikely to have been affected by any immediate effect on antipredator behaviour caused by transplanting fish from high $CO_2$ conditions in the laboratory to ambient $CO_2$ conditions in their natural habitat. Nevertheless, fish in the field studies also remained in control conditions for a much longer period of time (several days) compared with the laboratory tests conducted here (minutes) and further experiments of longer duration would be required to ascertain any differences in control and high $CO_2$ conditions that might accrue over a longer period of time.

In contrast to the absence of a test water effect on antipredator behaviours, we found that rearing treatment water had a highly significant effect on antipredator behaviour. Consistent with previous studies (*Dixson, Munday & Jones, 2010*; *Munday et al., 2010*), larval clownfish reared in high $CO_2$ exhibited a reversal from strongly avoiding predator cue (<10% of time in predator cue in the flume) to a strong attraction to predator cue (nearly 80% of time in predator cue in the flume). Similarly, larval damselfish exposed to high $CO_2$ for 4 days exhibited a slower response to a threat stimulus and a slower escape speed compared with fish kept in ambient control water. There was also a tendency for a shorter distance travelled. These findings are consistent with changes in escape responses that have has been reported previously (*Allan et al., 2013*; *Allan et al., 2014*) and provide further confirmatory evidence for the effects of high $CO_2$ on antipredator behaviour in larval reef fishes. The differences in escape speed between high $CO_2$ and control fish were relatively small; however, when combined with a much longer latency to respond (>30% increase in time to respond) these changes in kinematic responses would likely have a detrimental effect on the probability of a prey fish escaping a predator attack.

Altered behaviour of fish in a high $CO_2$ environment appears to be due to the effects of acid-base regulation on the function of GABA-A receptors (*Nilsson et al., 2012*; *Hamilton, Holcombe & Tresguerres, 2014*; *Heuer & Grosell, 2014*). Fish prevent plasma and tissue acidosis in a high $CO_2$ environment by accumulating $HCO_3^-$ in exchange for $Cl^-$ (*Brauner & Baker, 2009*). The GABA-A receptor is a gated ion channel that conducts $HCO_3^-$ and $Cl^-$ when activated by the neurotransmitter GABA. The altered concentration of $HCO_3^-$ and $Cl^-$ in fish exposed to high $CO_2$ could decrease or reverse the flow of these ions through the GABA-A receptor, causing a reversal in receptor polarization (*Heuer & Grosell, 2014*). Evidence for the role of GABA-A receptors in behavioural impairment of fish in high $CO_2$ has come from studies showing that treatment with gabazine, a GABA antagonist, reverses the effects of high $CO_2$ on behaviour (*Nilsson et al., 2012*; *Chivers et al., 2014*; *Chung et al., 2014*; *Lai, Jutfelt & Nilsson, 2015*; *Ou et al., 2015*; *Regan et al., 2016*). In the first of these studies, *Nilsson et al. (2012)* found that gabazine reversed the impaired response to predator odour exhibited by clownfish that had been reared in high $CO_2$. However, in that study gabazine was administered in control seawater and high-$CO_2$ reared fish were tested in control water. Therefore, the effects of gabazine in restoring the response of larval fishes to predator odour has not been previously tested in high-$CO_2$ conditions. Here we show that treatment with 4 mg $l^{-1}$ gabazine completely reverses the abnormal behavioural of

high-$CO_2$ fish to predator odour, restoring their natural aversion to predator odour. This contributes to a large number of independent studies pointing to a key role of GABA-A receptors in the behavioural impairment of fish exposed to high $CO_2$.

While we found no effects of using control or high $CO_2$ water in experimental trials to test the effects of high $CO_2$ on fish behaviour, *Sundin & Jutfelt (2016)* reported differences in relative lateralization of juvenile goldsinny wrasse when tested in control water versus high $CO_2$ water after 21 days in high $CO_2$. Their study did not include the reverse treatment, where control reared fish were tested in high $CO_2$ water, and there is the potential for learning effects in control fish because the same fish were tested on three different occasions, at days 9, 19 and 21. Nevertheless, that study does suggest that the response of some behavioural traits could be sensitive to differences in the test water used in behavioural trials. Further experiments with a broader range of behavioural traits and using experiments that are not complicated by the repeated testing of the same individuals are required to test this possibility. Furthermore, there are likely to be substantial species-specific differences in the response of fish to control or high $CO_2$ water. For example, *Jutfelt & Hedgärde (2013)* found that juvenile Atlantic cod were able to distinguish between low and high $CO_2$ water and exhibited a strong preference for low $CO_2$ water. Here, we only tested two species of damselfish, but other species might exhibit different responses. Indeed, there is considerable interspecific variation in the sensitivity of damselfishes to high $CO_2$ (*Ferrari et al., 2011a*), therefore, it is possible that there may also be species-specific differences in their antipredator responses depending on the test water used in the experimental arena.

Pioneering studies into the effects of high $CO_2$ on fish behaviour used control seawater in experimental trials due to logistical constraints and concerns about the possible effects of low pH on olfactory cues. Our results confirm the results of those initial studies by showing that test water (control of high $CO_2$) had no effect on the antipredator responses exhibited by fish reared in either control or high $CO_2$ conditions. Nevertheless, we do not advocate the continued use of control seawater for testing the behavioural effects of high $CO_2$ in marine organisms. Methods for treating seawater to replicate future $pCO_2$ levels are now sufficiently validated that it is possible to treat most experimental arenas with the respective treatment water that the animals have been reared in, and there may be some behavioural traits that are sensitive to different test water in the experimental trials. The most reliable and unambiguous results will likely come from using the respective treatment water in experimental trials testing the effects of high $CO_2$ on the behaviour of marine organisms. Future studies also need to consider the potential for adaptation of behavioural responses to projected future $CO_2$ levels (*Sunday et al., 2014*). Individual variation in behavioural tolerance to higher $CO_2$ levels (*Munday et al., 2012*) could provide the opportunity for selection of more tolerant genotypes and improved performance in a future high $CO_2$ environment.

## ACKNOWLEDGEMENTS

We thank staff at JCU's Marine and Aquaculture Facility and the Australian Museum's Lizard Island Research Station for logistical support.

### Funding

This research was funded by the ARC Centre of Excellence for Coral Reef Studies (P.L.M. & M.I.M.) and a Yulgilbar Foundation Fellowship (S.A.W.). The funders had no role in study design, data collection and analysis, decision to publish, or preparation of the manuscript.

### Grant Disclosures

The following grant information was disclosed by the authors:
The ARC Centre of Excellence for Coral Reef Studies (P.L.M. & M.I.M.).
Yulgilbar Foundation Fellowship (S.A.W.).

### Competing Interests

The authors declare there are no competing interestes

### Author Contributions

- Philip L. Munday conceived and designed the experiments, performed the experiments, analyzed the data, contributed reagents/materials/analysis tools, wrote the paper, prepared figures and/or tables, reviewed drafts of the paper.
- Megan J. Welch conceived and designed the experiments, performed the experiments, reviewed drafts of the paper.
- Bridie J.M. Allan conceived and designed the experiments, performed the experiments, analyzed the data, contributed reagents/materials/analysis tools, prepared figures and/or tables, reviewed drafts of the paper.
- Sue-Ann Watson conceived and designed the experiments, performed the experiments, analyzed the data, contributed reagents/materials/analysis tools, reviewed drafts of the paper.
- Shannon J. McMahon performed the experiments, reviewed drafts of the paper.
- Mark I. McCormick conceived and designed the experiments, contributed reagents/materials/analysis tools, reviewed drafts of the paper.

### Animal Ethics

The following information was supplied relating to ethical approvals (i.e., approving body and any reference numbers):

This research was conducted in accordance with James Cook University animal ethics guidelines and permits A1961, A1973 and A2080.

### Field Study Permissions

The following information was supplied relating to field study approvals (i.e., approving body and any reference numbers):

Fish were collected under permit G12/35117.1 from the Great Barrier Reef Marine Park Authority.

## Data Availability

Tropical Data Hub

http://dx.doi.org/10.4225/28/577DEDA8D0126.

## Supplemental Information

Supplemental information for this article can be found online at http://dx.doi.org/10.7717/peerj.2501#supplemental-information.

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
