# Peer review of "Effects of elevated CO2 on predator avoidance behaviour by reef fishes is not altered by experimental test water"

_PeerJ, doi:10.7717/peerj.2501_

## Round 0.1 · original submission · Minor Revisions

All three reviewers found your investigation to be sound and useful, but all also had suggestions for improvements. Their comments and reservations are clear and should be relatively easy to address. Two recommended minor revisions, one major revisions. Given the clarity and specific nature of the comments, I think you will be able to address these sufficiently. When you return your revision, please copy each review in its entirety and state beneath each concern or comment how you dealt with that comment. I anticipate being able to overview your responses, re-read the MS and make a decision without sending the MS out for another review.

Reviewer 1 ·

Basic reporting

This paper sets out to demonstrate the validity of one aspect of the methodology commonly used when examining CO2 effects on reef fish behavior, verifying that anti-predator responses are identical whether control or high CO2 water is used during experimental trials, regardless of whether larvae were raised under control of high CO2 conditions. While confirmatory in nature I believe this manuscript represents a small but necessary contribution to the literature and would be appropriate for publication in Peer J once the comments are addressed.

Experimental design

The husbandry and experimental protocols are both well established and used in the studies on which this manuscript it based so are appropriate. The experimental design and subsequent analysis are sufficient.

Validity of the findings

The rational for this study is clearly described.

Additional comments

Line 155: I assume both clutches were kept in control seawater until they hatched. Can water parameters during the egg stage have an effect on subsequent behavioral responses?

Line 183: The role of the dye test is not clear. From reading previous papers using flumes I understand that this is to ensure laminar flow and the absence of mixing but this should be stated.

Line 191: Please state why trials were discarded if temperature varied by more than 1 °C. I assume this is to prevent any effect of temperature shock on behavior.

Line 193: Was the experimenter also blinded to the test water used, both whether it was control or high CO2 and what flow contained predator odor?

Line 413: Your results seem to suggest that low pH does not decrease the potency of chemical predator cues, you could mention this.

Line 417: You only used one species of pomacentrid in each experiment and are assuming that these responses are representative of reef fishes. You should address the possibility of interspecific variation and mention that, while unlikely, fishes from other families could be affected by test water to some degree when using these or alternative experiments.

Reviewer 2 ·

Basic reporting

The article is well written, with only a few minor stylistic corrections and suggestions (see pdf).

There are a couple of places where the authors should more carefully and precisely define how their work is similar to or different than previous and related investigations (L119, L120) as well as more clearly describe previous findings and how they relate to the present work (L345, L414).

There are quite a number of places where the statistics are not reported in accordance with standard disciplinary practice (at least as this reviewer understands them; L304, 314, 315).

No link to the data repository is provided.

Experimental design

The experimental design is well suited to addressing the fundamental question posed by the investigators, and provides definitive data with respect to the role of test water on choice behavior.

It is less clear that the behavioral studies utilize sufficiently controlled methods, and there are some worrisome aspects. First, the description of the device used to provoke an escape response was confusing to this reviewer (L250)-this may be a place where a picture or diagram would be very helpful. Second, there are a variety of issues relating to the accuracy and precision of both the kinematic and measurements and analysis (L250, L 268) that need resolution before one has an adequate grasp of the uncertainty and utility of this data

Validity of the findings

The data is largely robust (particularly the choice data), generally placed into context appropriately and the conclusions valid. The authors generally do a good job of discussing the implications.

It is not clear that the statistical analysis of the choice data is appropriate for the experimental design; it does not seemingly incorporate the fact that the design as described uses the same fish for both test conditions; therefore it is a repeated measures design (L287), but is not described or presented as such.

This reviewer believes the claim that the present results validate the results of field trials is overstated (L361); there is a fundamental mismatch of time scales between the behavioral experiments presented here and that occurring in field trials, where fish are exposed to the ambient water conditions for different lengths of time. This must be acknowledged, and/or other information provided.

Annotated reviews are not available for download in order to protect the identity of reviewers who chose to remain anonymous.

Reviewer 3 ·

Basic reporting

This manuscript by Munday and colleagues is well written. It uses good, clear English. The Introduction describes the context and background well. The figures are clearly drawn and relevant. The raw data is provided. At times, the manuscript‘s finding are overstated, however

Experimental design

The experimental design employed by the study is well defined and relevant .
The gap in our understanding of how rearing conditions might (or might not) alter sensitivity to various sensory stimuli in the presence of elevated CO2 is addressed.
The investigation is conducted with technical prowess
The methodology is, in general, described in adequate detail although there is room for improvement (ex. more detail on water pH and chemistry would be highly desirable, as would more detail on the exact ages and exposure times of fish (these are relevant to neuronal development) would also be very helpful)

Validity of the findings

The results are valid although at times I believe overstated and oversimplified. II list a few that should be addressed. I recognize that many of these attributes are common to this discipline.
1) I do not object to the premise of the experiment that doubling water CO2 reflects the nature of the threat posed by climate change. I do find it misleading that it is not even mentioned that such changes are not occurring instantaneously but over decades and what could be 10’s generations of these fishes – easily perhaps enough time for evolutionary change to occur.
2) Many thousands of fish live on corals from many dozens of families. These fishes have very different genetics, physiologies, ecologies, etc., yet only a handful of these have been tested. This manuscript only tests fish from one family. These cannot be said to represent: “larval and juvenile coral fish” only fish of that family.
3) It is often stated that olfactory cues are being tested. Olfactory cues are stimuli detected by the olfactory sense but fish have several chemical senses (smell, taste, common chemical sense) and I am only aware of handful of studies that examined the roles of these systems in detecting predatory and alarm cues and all of these were in freshwater. Best to simply say “chemical cues.” (or justify)
4) It is implied/ stated that predatory cues are universal. None have been identified, at least definitively using modern biochemical techniques coupled with behavioral assays (Wisenden 2015) so this cannot be said. Best just say “washings of a predatory fish” (or similar).
5) This discussion asserts that chemical alarm cues from reef fishes degrade much more quickly in high C02 water…” However, chemical alarm cues have also never been definitively identified (Sorensen & Wisenden 2015 Fish pheromones and related cues); this has not been measured and should not be stated. Best remove this statement or justify it.

Additional comments

The abstract is overly long
Some the wording could be improved as mentioned above. The word “control” is often undefined. Its not really a control as nothing was blown into it. I’d prefer the term “low CO2 water”

It never really stated what the ecological relevance of background rearing levels of CO2 might be. Is there any??

---

## Round 0.2 · accepted · Accept

I judge your revisions to have adequately addressed the concerns raised by reviewers and am happy to accept the MS.